# Research on Preparation Methods of Carbon Nanomaterials Based on Self-Assembly of Carbon Quantum Dots

**DOI:** 10.3390/molecules27051690

**Published:** 2022-03-04

**Authors:** Xiaoqi Gao, Lei Wang, Chao Sun, Nan Zhou

**Affiliations:** Department of Applied Chemistry, Northeast Agricultural University, Harbin 150030, China; 18840614060@163.com (X.G.); wangluxurious@163.com (L.W.); sscc961027@163.com (C.S.)

**Keywords:** carbon quantum dots, self-assembly, graphene-like nano-carbon films, bifunctional magnetic-fluorescent composite, electrochemistry

## Abstract

Here, based on self-assembly of carbon quantum dots (CDs), an innovative method to prepare nanomaterials under the action of a metal catalyst was presented. CDs were synthesized by a one-step hydrothermal method with citric acid (CA) as the carbon source, ethylenediamine (EDA) as the passivator and FeSO_4_•7H_2_O as the pre-catalyst. In the experiment, it was found that the nano-carbon films with a graphene-like structure were formed on the surface of the solution. The structure of the films was studied by high-resolution transmission electron microscopy (HRTEM), Fourier transform infrared (FT-IR), etc. The results demonstrated that the films were formed by the self-assembly of CDs under the action of the gas–liquid interface template and the metal catalyst. Meanwhile, the electrochemical performance of the films was evaluated by linear cyclic voltammetry (CV) and galvanostatic charge discharge (GOD) tests. In addition, the bulk solution could be further reacted and self-assembled by reflux to form a bifunctional magnetic–fluorescent composite material. Characterizations such as X-ray diffractometer (XRD), fluorescence spectra (FL), vibrating sample magnetometer (VSM), etc. revealed that it was a composite of superparamagnetic γ-Fe_2_O_3_ and CDs. The results showed that self-assembly of CDs is a novel and effective method for preparing new carbon nanomaterials.

## 1. Introduction

In recent years, carbon nanomaterials composed entirely of sp^2^ bonded graphitic carbon, including zero-dimensional fullerenes, one-dimensional carbon nanotubes (CNTs), and two-dimensional graphene, have attracted particular attention due to their unique structural and physical properties. Among these materials, graphene, a monolayer graphite, is one single-atom thick, has presented with excellent conductivity, a large surface area, good mechanical properties, cost effectiveness, and feasibility toward chemical modifications, and it has already been widely applied in diverse fields, including energy storage device [1,2], biomedical science [3], barrier Polymers [4], and adsorption of gases [5].

On the nanoscale, the electronic structure is strongly related to the dimension and size of the specific object. For example, CDs have bandgaps, whereas graphene usually does not. In view of the pronounced quantum confinement effect, CDs have shown excellent properties, such as high aqueous solubility, high resistance to photo bleaching, strong and tunable photoluminescence, good electrical conductivity, chemical stability and benignity. Therefore, CDs have been widely used in various fields, such as bioimaging [6,7], sensors [8], drug delivery [9,10], chemiluminescence [11,12] and energy storage [13]. Synthetic approaches for CDs can be classified into two categories, viz. top-down and bottom-up methods. Among them, the hydrothermal method has been widely used in the preparation due to its simplicity and low cost [14,15]. In particular, CDs prepared hydrothermally, using citric acid (CA) as a carbon source and organic amine compounds as passivators, have exhibited excellent photoluminescence properties. The typical structure of CDs was treated as a core–shell model, consisting of a carbon core with a graphite lattice structure and a shell made of various surface functional groups, such as amino, carboxyl, and amide groups. The photoluminescence characteristics were jointly determined by the two parts of the structure [16,17]. Specifically, there were two different theories about the reaction mechanism of such CDs. First, non-conjugated molecules (such as CA) could form ammonium salts with ammonia compounds. Second, the ammonium salts were further cyclized and aromatized by direct amidation reaction to generate a conjugated sp^2^ structure, which contained heterocyclic compounds such as imidazole, pyridine, etc. Finally, on this basis, it was further carbonized to form CDs [16,18,19]. Some researchers also believed that CDs prepared from CA and ethylenediamine (EDA) could form amorphous linear polymers or cross-linked polymers via amidation reaction due to the abundant carboxylic acid and amine groups on the surface and then carbonize to form the carbon core on this basis [19,20]. The size could be well controlled below 10 nm (usually between 2 and 5 nm) without adding surfactants or other size-limiting techniques. Furthermore, CDs themselves would not grow up, and the amino and carboxyl groups between CDs would hardly undergo amidation reaction; that is, there would be no polymerization reaction between CDs. Guo et al. [21] tried to improve the optical properties of the material by coupling CDs together through amide bonds. Wei et al. [22] used the simple CDs as a template to induce hydrothermal self-assembly of ultrathin Ni(OH)_2_ nanosheets. However, no attempts have been made to use CDs as basic units to construct carbon nanomaterials by self-assembly.

In the field of organic synthesis, catalysts were usually added to the system to improve the efficiency of the amidation reaction, which was also called the catalytic direct amidation reaction. The main function of the catalyst was to reduce the salt-forming reaction and enhance the electrophilicity of the carboxy group as much as possible, such that the nucleophilic addition-α elimination reaction could proceed smoothly [23,24,25,26,27,28]. Transition metal catalysts such as iron salts could be used as electrophiles to provide empty orbitals in the reaction, form complexes with the substrate, reduce the activation energy of the reaction, and catalyze the progress of the reaction. Basavaprabhu et al. [23] chose phenylacetic acid and aniline as model substrates and investigated the effects of direct amidation catalyzed by different metal iron salts. They reported that FeCl_3_ had the best catalytic effect, and the iron coordination increased the electrophilicity of the carbonyl group, thereby triggering the nucleophilic attack of the amine.

Inspired by the catalytic direct amidation reaction in organic synthesis, we envisaged introducing the catalyst to promote the amidation reaction between CDs and construct new carbon materials through the self-assembly of CDs. In this work, two-dimensional carbon nanofilms and bifunctional magnetic–fluorescent composites were prepared by a one-step hydrothermal method and hydration method, with CA as the carbon source, EDA as the passivator and FeSO_4_•7H_2_O as the catalyst. We analyzed the structure and put forward the possible synthesis mechanism of nanomaterials and initially tested the relevant properties.

## 2. Results and Discussion

### 2.1. Structure of GN-1.5

As shown in Figure 1a, when only citric acid and ethylenediamine were used as reactants, hydrothermal treatment resulted in a reddish-brown colidal liquid, and no film appeared at the gas–liquid interface. When FeSO_4_•7H_2_O (acting as a catalyst) was added, the metallic luster films (called GN-1.5) were produced under the same process conditions. The morphology and size of GN-1.5 were obtained from HRTEM images. As shown in Figure 1, both monolayer (in Figure 1c) and multilayer (in Figure 1b,d) transparent wafers were presented in the transparent wafer GN-1.5. Figure 1b presents the features of multilayer sheets and regular edges of GN-1.5. As can be seen in Figure 1c,d, GN-1.5 had the characteristics of bending and folding, which might have been caused by the superposition of graphene-like sheets or the coiling of the edges. It can be seen from Figure 1e,f that GN-1.5 was assembled by circular structures. The lattice spacing of circular structures was 0.21 nm, which was a typical characteristic lattice spacing of CDs. Moreover, the connected part between the CDs also presented a complete lattice structure. The well-resolved lattice fringe of 0.21 nm was the result of the d-spacing of graphene (100 facet), which corresponded to the hexagonal lattice [29,30]. As shown in Figure 1g, in order to reduce errors and facilitate repeated verification, five rectangular regions were selected for fast Fourier transform (FFT), and all of them presented a hexagonal diffraction pattern, which corresponded to the lattice fringe spacing [30].

The XRD spectrum revealed the high crystallinity of GN-1.5, as shown in Figure 2a. The diffraction peaks appearing at 35.0° and 62.7° corresponded to the (111) and (440) crystal planes of Fe(OH)_3_ (JCPDF card no. 22-0346). The diffraction peak appearing at 26.5° corresponded to the (002) crystal plane of graphene (JCPDF card no. 41-1487).

FT-IR spectroscopy was conducted to analyze the structure and surface functional groups of GN-1.5, as shown in Figure 2b. In regard to GN-1.5, O-H and N-H (3271cm^−1^, a broad band), C-NH [31]/CH_2_ [17] (1396 cm^−1^), C-NH-C [32] (1128 cm^−1^) and the skeleton vibration of C-N [33] (1025 cm^−1^) were observed. The peak at 1616 cm^−1^ corresponded to the C=C stretching vibration and C=N stretching vibration of the aromatic ring, which confirmed the presence of carbon atoms in sp^2^ graphene and the nitrogen doping of graphene. In particular, the C=O stretching vibration of amide also appeared in this wave number [34]. This might have been because that the amide bond connected to the sp^2^ conjugated structure, and the conjugation effect produced caused the blue shift of the C=O wave number. The peaks at 711 and 484 cm^−1^ were the characteristic peaks of Fe and Fe-OH, respectively [35]. These results indicated that iron hydroxide and nitrogen-doped graphene-like were successfully combined.

The EDS spectrum (in Figure 2c) showed that the surface of GN-1.5 contained five elements: C, N, O, Fe, and S. Among them, the contents of O, Fe and S were relatively high, while the contents of C and N were relatively low. Combined with the XRD and FT-IR analysis results, it was speculated that iron hydroxide was formed and some iron sulfate salts were attached to GN-1.5, which made O, Fe, and S inevitably cover the two elements of C and N.

### 2.2. Electrochemical Property Study for GN-1.5

The potential range of 0.1 to 0.4 V was selected. As can be seen in Figure 3a, the CV curves exhibited typical pseudo-capacitive behavior, showing the volt–ampere characteristics of a pair of obvious redox peaks. With the increase in the scan rate, the shapes of the CV curves were almost unchanged, indicating that the composite material was a good electronic conductor with a small equivalent series resistance. With the increase in the scan rate, the anode peak and cathode peak did not shift, indicating that the internal resistance of the GN-1.5 electrode was relatively small.

The GOD measurements were performed when the charging voltage was 0 V and the discharging voltage was 0.4 V. As shown in Figure 3b, the GOD curves of GN-1.5 electrode displayed a sudden potential drop followed by a slow potential decay, which was the inherent characteristic of pseudocapacitative materials. The specific capacitance from the GOD measurements can be evaluated from the following Equation (1):(1)$$C_{S}=\frac{I\times t}{m\times V}$$
where *C_s_* is the specific capacitance (F g^−1^), *I* is the charge–discharge current (A), *t* is the charge–discharge time (s), *m* is the mass of the active material in the electrode (g) and *V* is the charge–discharge potential range (V). With the increase in current density, the specific capacitance gradually decreased to 218.00, 176.64, 160.83, 126.66 and 110.83 F g^−1^ at 2, 4, 5, 8 and 10 A g^−1^, respectively, which was probably because some parts of the surface of GN-1.5 were inaccessible at high charge and discharge rate. However, the specific capacitance of GN-1.5 was still as high as 110.83 F g^−1^ even at a high current density of 10 A g^−1^. As shown in Figure 3c. The sample GN-1.5 shows a much better cycling stability, with 83.92% retention of the initial capacity after 2000 cycles at 0.5 A g^−1^.The results showed that the films were expected to be applied in electrochemical fields such as supercapacitors.

### 2.3. Self-Assembly Mechanism of the Graphene-like Monolayer Film

The basic premise for constructing high-dimensional (one-dimensional, two-dimensional or three-dimensional) nanostructures based on self-assembly of zero-dimensional materials such as nanocrystals is the asymmetry of the structure [36,37,38]. This is the physical basis for the faster growth of nanocrystals in one dimension. CDs, with anisotropic geometry, are made up of a carbon core with a graphite lattice structure and various surface functional groups such as amino, carboxyl and amide groups. The shape of CDs is similar to an ellipsoid, which means smaller dimensions perpendicular to the direction of the layered structure [16,17,37,38]. First, FeSO_4_•7H_2_O is ionized in water to produce Fe^2+^. The presence of the pre-catalyst catalyst (called Fe^2+^) is conducive to catalyzing the direct amidation reaction between CA and EDA and improving the catalytic efficiency [23]. The coordination of Fe^2+^ with the carbonyl group in CA increases the electrophilicity of carbon in the carbonyl group, thereby triggering the nucleophilic attack of EDA to form CDs. At the same time, hydroxyl radicals generated by the hydrolysis of EDA may oxidize Fe^2+^ to Fe^3+^. Fe^3+^ is adsorbed on the surface functional groups of CDs, which can significantly reduce the growth potential energy of CDs and further limit the size of CDs [39].

The surface structure of CDs is similar to polycyclic aromatic hydrocarbon (PAH), and the hydrophilic functional groups are mainly concentrated on the edge of the sheet structure. The direction perpendicular to the layer is the hydrophobic surface, while the sheet edge parallel to the layer contains a large number of hydrophilic groups as the hydrophilic surface. When CDs move to the gas–liquid interface, the hydrophobic surface and the gas–liquid interface are arranged in parallel. Under the catalysis of Fe^3+^ and guided by the layered graphite structure, the amide bonds are first formed between CDs, and then CDs undergo dehydration for self-assembly [23]. As the hydrophobic interface expands, the formed sheet-like film can no longer enter the body of the solution but floats above the liquid surface. Due to the guidance effect of graphite layered structure, which is similar to the role of seed crystal in the growth process of a single crystal, the connecting parts between CDs present a complete lattice structure, making the nanofilm present a lattice similar to that of graphene. However, limited by the thickness of CDs (2–3 nm), the thickness of the single-layer film is equivalent to 6–9 layers of graphene. In the body of the solution, Fe^3+^ is combined with the hydroxyl radicals generated by the hydrolyzation of ethylenediamine to form ferric hydroxide and adheres to the surface of the film.

### 2.4. The Structure and Properties of MFN

To analyze the chemical structure of MFN-1.5, we characterized them by XPS. As shown in Figure 4a–e, the XPS spectra showed four major binding energy peaks: 297.98, 544.98, 739.98, and 410.03 eV, which corresponded to the binding peaks of C 1s (49.65%), O 1s (33.35%), Fe 2p (8.79%) and N 1s (7.18%). The binding energy peak of S 2p at 175.03 eV cannot be separated due to its extremely low sulfur content, accounting for 1.04% of the total. Specifically, the C 1s spectra (in Figure 4b) showed peaks at 284.8, 288.2, 286.2 and 287.0 eV, which were attributed to C-C/C=C [40], C=O, C-N and C-O, respectively. The fitted peaks at binding energies of 399.8, 401.7 and 401.0 eV in the N 1s region (Figure 4c) originated from C-N-C [41], N-H [42] and C-NH_2_, respectively. The XPS spectrum of O 1s is presented in Figure 4e, which was divided into two peaks located at 531.1 and 529.8 eV, corresponding to C=O and Fe-O, respectively. In Figure 4e, two main peaks at 710.6 and 724.6 eV were separate for Fe 2p_3/2_ and Fe 2p_1/2_, which matched with the reported value for Fe_2_O_3_. The binding energies of 718.8 and 733.4 eV corresponded to the Fe 2p_3/2_ and Fe 2p_1/2_ satellites of Fe_2_O_3_, respectively [43,44,45]. No Fe^2+^ peaks (ca. 716 eV) were observed in the spectrum, confirming the high purity of the Fe_2_O_3_ crystalline structure without the interference of Fe_3_O_4_ [44].

To further study the crystal structure and phase composition of MFN-1.5, the synthesized product MFN-1.5 was analyzed by XRD, as shown in Figure 4f. The well-resolved diffraction lines in the pattern could be represented by γ-Fe_2_O_3_ (JCPDF card no. 39-1346). The diffraction peaks of γ-Fe_2_O_3_ appeared at 18.3°, 30.2°, 35.6°, 43.2°, 57.2° and 62.9°, which could be perfectly consistent with (111), (220), (311), (400), (511) and (440) crystal planes. No additional signals of possible impurities were detected, such as the characteristic peaks of β-FeOOH or α-Fe_2_O_3_, indicating that the product had high purity. No diffraction peaks of carbon were observed, which indicated that the samples were pure γ-Fe_2_O_3_, and the carbon produced by the reaction was amorphous carbon.

FT-IR spectroscopy was conducted to analyze the structure and surface functional groups of MFN, as shown in Figure 5a. In regard to MFN-0, the strong peaks at 2926, 1653, 1573 and 1388 cm^−1^ were attributed to the C-H stretching vibration of alkane, C=O stretching vibration in amide [46], C=C stretching vibration in the aromatic ring [46] and C-NH bending vibration [31]. The peak at 1055 cm^−1^ belonged to the C-N framework vibration [33]. The wide band with the center of 3278 cm^−1^ indicated the presence of -OH and N-H. The peaks at 3396 and 3278 cm^−1^ could be attributed to the O-H/N-H stretching vibration of the amino group and the hydroxyl group. These data indicated that there were amino groups on the surface of MFN-0 in addition to carboxyl and hydroxyl groups. The peak positions of MFN-1.5 and MFN-0 were basically the same (above 1000 cm^−1^), and they had good hydrophilicity and stability in water systems.

Differently, the C=C stretching vibration peak of MFN-1.5 overlapped with C=O stretching vibration at 1631 cm^−1^ [34]. The peaks at 635 and 596 cm^−1^ belonged to the Fe-O bond vibration of iron oxide. The peak at 850 cm^−1^ was attributed to the residual peak of maghemite. These three peaks confirmed the formation of γ-Fe_2_O_3_ [47,48,49]. At the same time, it was found that the bending vibration peak intensity of C-NH of MFN-1.5 was significantly stronger than that of MFN-0, which proved that iron ions effectively catalyzed the formation of amide bonds.

The optical properties of the MFN were characterized by means of UV–Vis and fluorescence emission spectra, as shown in Figure 5b,c. The UV–Vis absorption measurement showed two characteristic absorption peaks at 239 and 343 nm, which were attributed to the π^_^π* transition of aromatic sp^2^ domains from the carbon core and the n^_^π* transition of C=O, respectively [50]. The absorption peak at 343 nm came from the capture of the excited state energy by the n-state on the surface, resulting in a strong blue light emission [51]. In addition, we found that the absorption value of MFN-1.5 at 239 nm was slightly higher than that of MFN-0, when the UV–Vis absorption value at 343 nm was controlled to be the same.

The fluorescence emission peaks of MFN-0 and MFN-1.5 at the maximum excitation wavelength of 343 nm was located at 439 nm in Figure 5c. Both had broad emission spectra and exhibited strong blue emission under 365 nm ultraviolet lamp. We found that the fluorescence intensity of MFN-1.5 was significantly lower than that of MFN-0 when the UV–Vis absorption value was the same at 343 nm. It was speculated that the decrease in fluorescence intensity may be caused by the adsorption of a large amount of iron ions.

The magnetic properties of MFN-1.5 at room temperature were studied, as shown in Figure 5d. An appropriate amount of ultrapure water was added to the petri dish, and then MFN-1.5 was added to the dish. When the magnet was placed on the left (inset (d-1) in Figure 5), MFN-1.5 was observed to move toward the magnet under the irradiation of an ultraviolet lamp with the selected wavelength of 365 nm. When the magnet was placed under the dish (inset (d-2) in Figure 5), the diffusion of MFN-1.5 was limited to the area above the rectangular magnet, which avoided overflowing of the edge. These phenomena proved the successful combination of MFN-1.5 magnetism and fluorescence. The hysteresis loop showed that MFN-1.5 had superparamagnetic properties and that the saturation magnetization was 11.14 emu/g in Figure 5d.

The morphology and size of MFN-1.5 were obtained from HRTEM images. As shown in Figure 6a,b, MFN-1.5 was amorphous and gel-like in shape, with overlapping accumulations. Many small black dots were attached to the amorphous colloidal substance in Figure 6c. These dots were spherical in shape and uniformly dispersed, with an average diameter of 2.83 nm. According to the above results, it could be proven that the dots were superparamagnetic γ-Fe_2_O_3_ nanoparticles.

As shown in Figure 6d, the HRTEM image showed that MFN-1.5 had no clear lattice fringes, which was attributed to the amorphous substance. The inability to form graphene-like films such as GN-1.5 was related to the template effect of the gas–liquid interface. CDs were anisotropic [37,38]. When CDs moved to the gas–liquid interface, the hydrophobic surface of CDs was aligned parallel to the gas–liquid interface. Under the catalysis of iron ions and guided by the layered graphite structure, the amide bonds were first formed between CDs, and then CDs dehydrated and self-assembled into a graphene-like film. However, during the formation of MFN-1.5, the first step hydrothermal method did not directly combine magnetism with fluorescence. The magnetic particles gathered at the bottom of the reactor and the supernatant only had fluorescence. The second step of reflux provided the reaction system with a suitable temperature and sufficient oxygen, which not only redissolved the large magnetic particles, but also made the unreacted raw materials in the solution body form an amorphous colloidal substance under the action of the catalyst. Metal oxides and CDs were stuck together by an amorphous colloidal substance to form a magnetic fluorescent dual-functional composite.

## 3. Materials and Methods

### 3.1. Materials

FeSO_4_•7H_2_O was purchased from YongDa Chemical Reagent Co., Ltd. (Tianjin, China). Citric acid (CA), KBr and ethanol absolute were purchased from Aladdin Chemical Reagent Co., Ltd. (Shanghai, China). Acetylene black and polytetrafluoroethylene (PTFE) emulsion were purchased from Shengernuo Technology Co., Ltd. (Suzhou, China). All the reagents mentioned above were of analytical grade, expect for the spectrally pure KBr, and were used without further purification. Ultrapure water was used throughout the experiments.

### 3.2. Synthesis of the Graphene-like Nano-Carbon Films

First, 5.2 mmol CA was dissolve with 0 and 5.4 mmol FeSO_4_•7H_2_O in 29 mL of ultrapure water, respectively, and then 1 mL of ethylenediamine was added and mixed well. Next, the mixture was transferred into a 50 mL Teflon-lined stainless-steel autoclave and heated for 5 h at 180 °C. After the reaction finished, the obtained product was naturally cooled to room temperature and left to stand for 12 h. The samples were denoted as GN-0 and GN-1.5. Without the catalyst of FeSO_4_•7H_2_O, no films appeared at the gas–liquid interface of the reactor. When 5.4 mmol FeSO_4_•7H_2_O was added, the metallic luster films (called GN-1.5) were produced. Then, the films were repeatedly washed with ultrapure water and ethanol until there was no blue fluorescence on the surface. Finally, they were stored in a 3 mL centrifuge tube filled with ethanol absolute.

### 3.3. Synthesis of the Composite of γ-Fe_2_O_3_ and CDs

This preparation procedure was exactly the same as that of graphene-like films, except that the product in the autoclave was naturally cooled to room temperature, and the solution and magnetic solid precipitate in the autoclave were quickly transferred to a three-necked flask. The thermometer was placed on the left branch of the three-necked flask (above the liquid level), the condenser was connected to the middle of the flask, and the right branch was sealed with a rubber stopper. The stirring button was turned to the maximum value, and the temperature of the system was stably controlled at 98 °C and refluxed for two hours. Next, the contents of the flask were transferred to a 50 mL transparent centrifuge tube and cooled naturally to room temperature. They were then centrifuged at 10,000 rpm for ten minutes by utilizing a high-speed centrifuge. Finally, the supernatant was taken out by a disposable dropper, and the samples were denoted as MFN-0 and MFN-1.5. In order to remove other small organic impurities and most of the water, all prepared MFN were repeatedly extracted and purified with n-butanol. Finally, the purified MFN were placed in the dark at room temperature for further experiments.

### 3.4. Characterization

High-resolution transmission electron microscopy (HRTEM) was performed with a JEOL JEM 2100F microscope at an accelerating voltage of 200 kV. The energy dispersive X-ray spectrometer (EDS) was equipped to obtain the mapping spectrum. X-ray photoelectron spectroscopy (XPS) was performed with a Thermo Scientific Escalab 250Xi (Thermo Fisher Scientific, Waltham, MA, USA). Fourier transform infrared (FT-IR) spectroscopy was conducted with a Magna-IR560 FT-IR spectrometer (Nicolet Co., Madison, WI, USA). UV–Vis absorption spectroscopy was conducted on a UV-2550 spectrophotometer (Shimadzu, Kratos, Japan). Fluorescence spectra (FL) of the MFN were recorded with an LS-55 fluorescence spectrometer (PerkinElmer, Boston, MA, USA). Both GN-1.5 and MFN-1.5 samples were identified by X-ray diffraction (XRD, Bruker D8 Advance). The scanning angle was 5–90°, and the scanning rate was 2°/min. The room temperature magnetic behavior of the sample (called MFN-1.5) was investigated using a vibrating sample magnetometer (VSM, Quantum Design-PPMS).

### 3.5. Electrochemical Measurements

The CDs on the surface of GN-1.5 were cleaned by ethanol absolute, and then the ethanol was cleaned by ultrapure water. The working electrode was prepared as follows: First, 85 wt% of active material and 15 wt% of acetylene black were ground into a uniform black powder in a mortar. Then, an appropriate amount of PTFE emulsion and a few drops of ethanol were added into the mixed-powder and sonicated to form the slurry. After mixing together, the resulting slurry was coated onto nickel foams (about 2 mg of active material and with a geometric surface area of about 1 cm^2^). Finally, the electrodes were dried for 8 h at 60 °C in air and pressed at 10 MPa.

All electrochemical measurements were carried out on an electrochemical working station (CS35OH, Wuhan, China) in a three-electrode system in 3 M KOH aqueous electrolyte at room temperature. The saturated calomel electrode (SCE) and Pt electrode were utilized as the reference and counter electrode, respectively. Cyclic voltammetry (CV) and galvanostatic charge/discharge (GCD) were used to determine the electrochemical performance of GN-1.5.

## 4. Conclusions

Using citric acid as a raw material and ethylenediamine as a nitrogen source, CDs were self-assembled at the gas–liquid interface to form the composite of Fe(OH)_3_ and graphene-like. The thickness of the unilaminar film was about 6–9 layers of graphene, which provided a new idea for the synthesis of high-quality multilayer graphene-like under simple and mild conditions. The CV as well as GOD studies of the composite of Fe(OH)_3_ and graphene-like demonstrated that the films delivered a specific capacitance of 218.00 F g^−1^ at the current density of 2 A g^−1^. The results showed that the composite of Fe(OH)_3_ and graphene-like held great potential as high-performance electrode materials for supercapacitors. The bulk solution successfully combined magnetism with fluorescence via hydrothermal and reflux to prepare the water-soluble, superparamagnetic γ-Fe_2_O_3_-CDs composite. Briefly, under the action of a catalyst, self-assembly with carbon quantum dots as basic structural units through direct amidation reaction is a novel and effective method for preparing carbon nanomaterials and composite functional materials.

## Figures and Tables

**Figure 1 molecules-27-01690-f001:**
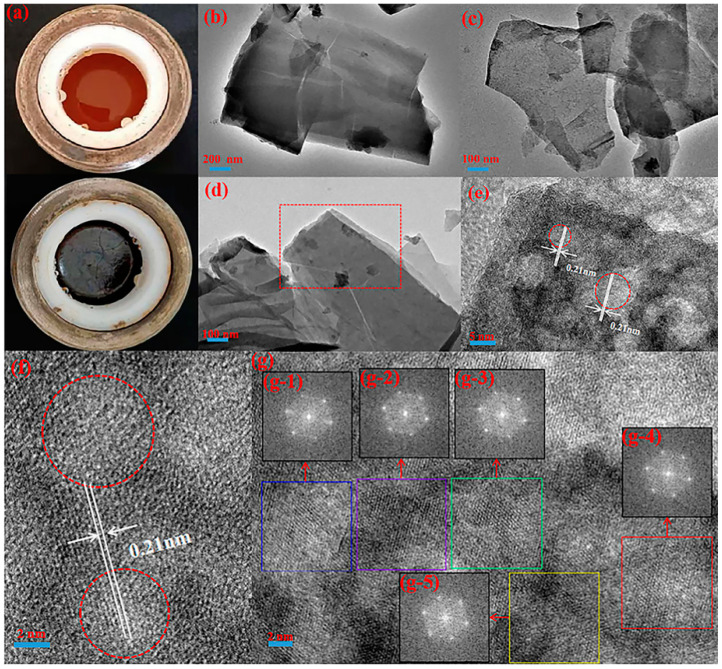
(**a**) Reaction autoclave diagrams of GN-0 and GN-1.5 in the control group; (**b**–**d**) HRTEM images of GN-1.5 at different magnification; (**e**–**g**) The lattice structure of the GN-1.5 obtained from HRTEM image; (**e**) the lattice diagram corresponding to the red region in (**d**–**f**) are the edge and internal lattice diagrams of GN-1.5, respectively; inserts (**g**-**1**), (**g**-**2**), (**g**-**3**), (**g**-**4**), and (**g**-**5**) fast Fourier transform (FFT) patterns of the rectangular region correspond to the blue, purple, green, red, and yellow boxes in (**g**), respectively.

**Figure 2 molecules-27-01690-f002:**
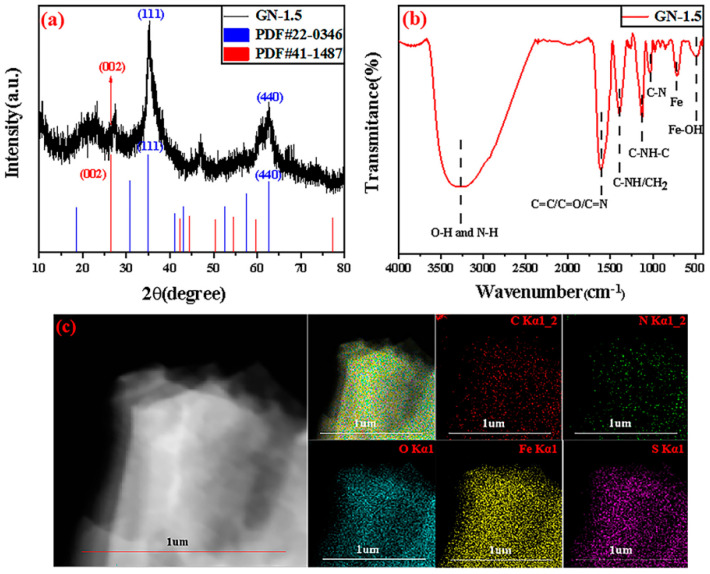
(**a**) XRD pattern of GN-1.5; (**b**) FT-IR spectrum of GN-1.5; (**c**) EDS spectrum of GN-1.5.

**Figure 3 molecules-27-01690-f003:**
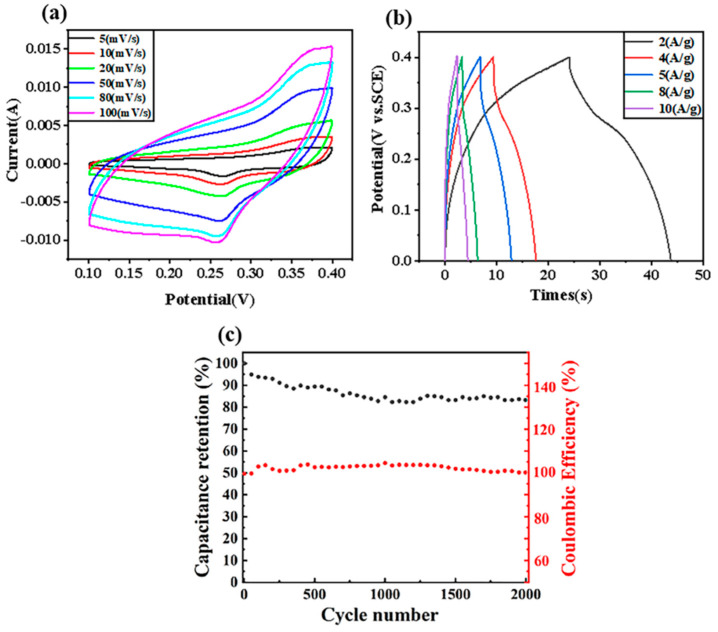
(**a**) The cyclic voltammetry (CV) curves of GN-1.5 electrode at scan rates of 5–100 mV s^−1^. (**b**) The galvanostatic charge discharge (GOD) curves of GN-1.5 electrode at current densities of 2–10 A g^−1^. (**c**) The cycling performances of GN-1.5 at 0.5 A g^−1^.

**Figure 4 molecules-27-01690-f004:**
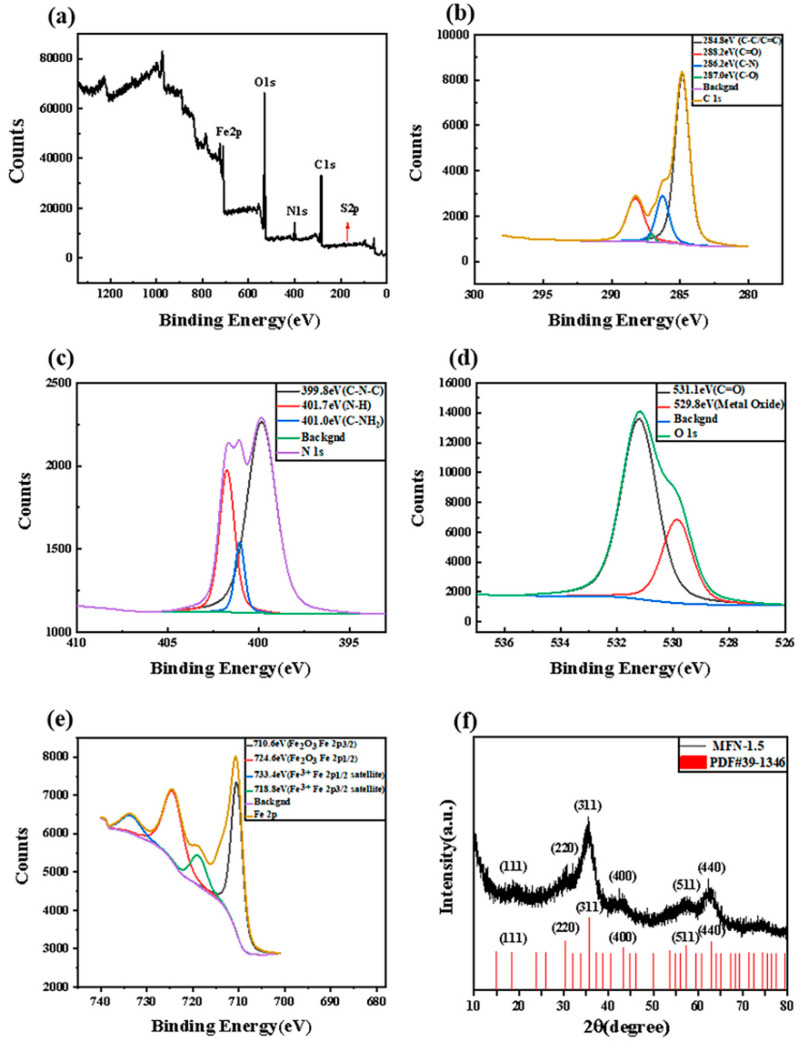
(**a**) The full-scale XPS spectrum of MFN-1.5; (**b**) C 1s spectrum; (**c**) N 1s spectrum; (**d**) O 1s spectrum; (**e**) Fe 2p spectrum; (**f**) XRD pattern of MFN-1.5.

**Figure 5 molecules-27-01690-f005:**
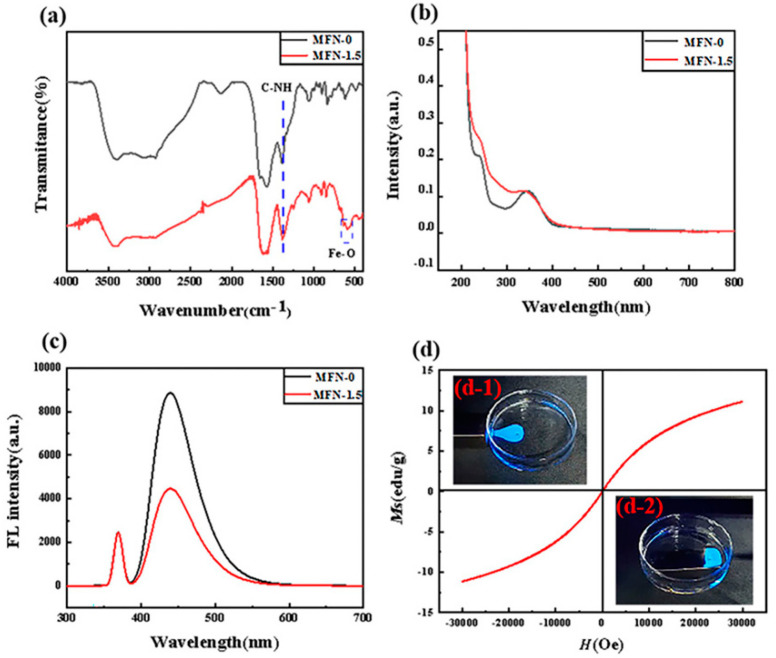
(**a**) FT-IR spectra of MFN-0 and MFN-1.5; (**b**) UV–Vis spectra of MFN-0 and MFN-1.5; (**c**) fluorescence emission spectra of MFN-0 and MFN-1.5; (**d**) magnetization hysteresis loop of MFN-1.5, inserts (**d**-**1**) and (**d**-**2**) state diagrams of MFN-1.5 after being attracted in ultrapure water when the magnet was placed on the left side and directly below the petri dish.

**Figure 6 molecules-27-01690-f006:**
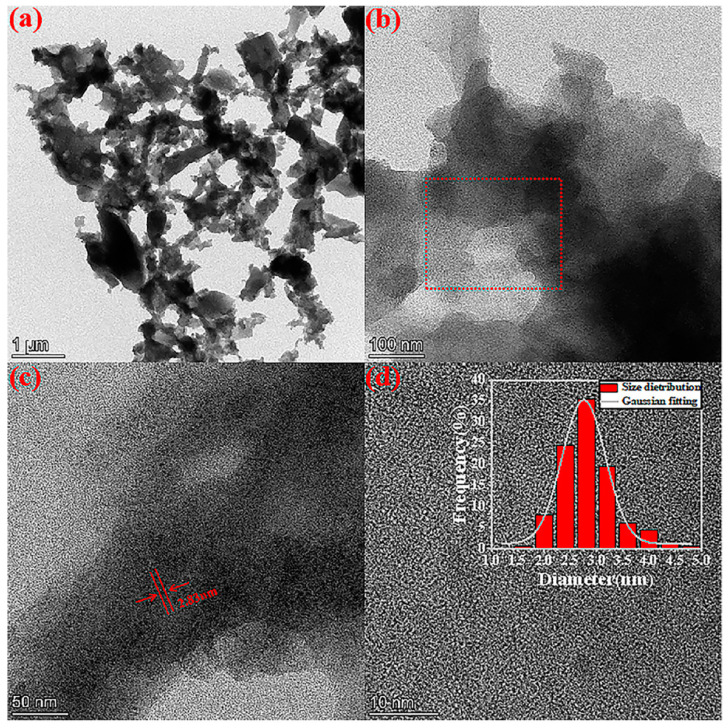
(**a**–**c**) HRTEM images of MFN-1.5 under different magnifications; (**c**) enlarged from the red rectangular area in (**b**); (**d**) the lattice diagram obtained from the HRTEM images; (insert) particle-size distributions of γ-Fe_2_O_3_. Two hundred particle sizes were counted.

## Data Availability

Not applicable.

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
