# Peer review of "Research on Preparation Methods of Carbon Nanomaterials Based on Self-Assembly of Carbon Quantum Dots"

_molecules, 2022, doi:10.3390/molecules27051690_

Round 1
Reviewer 1 Report
Corrections to minor methodological errors and major corrections in text editing
Check the English and the separation of words at the end of the lines for example max-imun, if you want to write separated you must maxi-mun or fer-rous.
In page 5 this phrase is not well written "than the dimension"
Check references 18, 23, 39,
10,000 rpm/min, that's wrong, only 10000 rpm must be written
maybe is most correct autoclave than kettle.
Author Response
Dear Editor,
Thank you very much for your attention and the reviewers’ comments on our manuscript entitled “Research on preparation methods of carbon nanomaterials based on self-assembly of carbon quantum dots” (ID: molecules-1565409). The comments are all valuable and very helpful for revising and improving our manuscript, as well as the important guiding significance to our researches.
We have checked and revised our manuscript carefully according to these comments, and the revised portions are marked in red in the manuscript. Enclosed please find the responses to the reviewers. We sincerely hope this manuscript will be finally acceptable to be published on Molecules.
Thank you and best regards.
Responses to reviewer’s comments are as follows:
Thank you for your positive suggestions. Please allow us to answer your concerns below.
Comment 1:Corrections to minor methodological errors and major corrections in text editing
Response: Thank you for the advice. We have revised this.
Comment 2:Check the English and the separation of words at the end of the lines for example max-imun, if you want to write separated you must maxi-mun or fer-rous.
Response: Thank you for the advice. We have revised this.(line 248)
Comment 3:In page 5 this phrase is not well written "than the dimension"
Response: Thank you for the advice. We have revised this.(line 165-line166)
Comment 4:Check references 18, 23, 39,
Response: Thank you for the advice. We have revised this.(line427,line445,line461)
Comment 5:10,000 rpm/min, that's wrong, only 10000 rpm must be written
Response: Thank you for the advice. We have revised this.(line 319)
Comment 6:maybe is most correct autoclave than kettle.
Response: Thank you for the advice. We have revised this.(line 311, line312, line 528, line 557)

Reviewer 2 Report
Though the hydrothermal synthesis of carbon dots from citric acid and ethylenediamine is not a brand new approach, the results and detailed analysis by Gao et al. are still useful for the community. Therefore, I suggest this manuscript to be published after a minor revision.
Detailed comments:
- It is not accurate to state “zero-dimensional carbon quantum dots (CDs) are structurally different from graphene, they have an electronic structure similar to that of graphene”. At nanoscale the electronic structure is strongly related to the dimension and size of the specific object. For example, the carbon dots have bandgaps, whereas a graphene usually does not.
- In the method sections, it would be better if the author can include the moles of each reagents.
- The author claimed FeSO4 as the catalysts. In my opinion, Fe2+ could be a pre-catalyst, and Fe3+ is the catalyst during hydrothermal synthesis. Indeed, Fe3+ is widely used for C-C coupling in bottom-up synthesis of nano-graphenes.
- The “Self-assembly mechanism of the graphene-like monolayer film” needs to be carefully polished. There are several misleading discussions. (1) Ferrous ions do not bind a “hydroxide radical”. Hydroxide radical is not a scientific name, and it should be hydroxyl radical. (2) Hydroxide radical may oxide Fe2+. (3) The authors should provide references to the claimed mechanism and also consider the effects from Fe3+.
Author Response
Dear Editor,
Thank you very much for your attention and the reviewers’ comments on our manuscript entitled “Research on preparation methods of carbon nanomaterials based on self-assembly of carbon quantum dots” (ID: molecules-1565409). The comments are all valuable and very helpful for revising and improving our manuscript, as well as the important guiding significance to our researches.
We have checked and revised our manuscript carefully according to these comments, and the revised portions are marked in red in the manuscript. Enclosed please find the responses to the reviewers. We sincerely hope this manuscript will be finally acceptable to be published on Molecules.
Thank you and best regards.
Responses to reviewer’s comments are as follows:
Thank you for your positive suggestions. Please allow us to answer your concerns below.
Comment 1:It is not accurate to state “zero-dimensional carbon quantum dots (CDs) are structurally different from graphene, they have an electronic structure similar to that of graphene”. At nanoscale the electronic structure is strongly related to the dimension and size of the specific object. For example, the carbon dots have bandgaps, whereas a graphene usually does not.
Response: Thank you for the advice. As you said, the electronic structure of CDs and graphene is different. Our experiments show that the junction has a complete lattice structure after CDs self-assemble to form a two-dimensional carbon material. The result of Fourier transform (FTT) and lattice fringe show that the crystal structure of the film is similar to multilayer graphene. So we infer that CDs and graphene have similar electronic structures. Our expression is wrong, and it has been revised in the corresponding position of the article. (Line 35)
Comment 2:In the method sections, it would be better if the author can include the moles of each reagents.
Response: Thank you for the advice. We have revised this. (Line 298, 304)
Comment 3:The author claimed FeSO4 as the catalysts. In my opinion, Fe2+ could be a pre-catalyst, and Fe3+ is the catalyst during hydrothermal synthesis. Indeed, Fe3+ is widely used for C-C coupling in bottom-up synthesis of nano-graphenes.
Response: Thank you for the advice. We have revised this. (Line 9, 166, 181,182 )
We found literature supporting that both divalent and trivalent iron ions can catalyze amide bond formation between phenylacetic acid and aniline. (Line 445-447)
- Basavaprabhu, B.;Muniyappa, K.; Panguluri, N. R.; Veladi, P.; Sureshbabu, V. A simple and greener approach for the amide bond formation employing FeCl3 as a catalyst. New. J. Chem. 2015, 39 (10), 7746-7749. DOI: 10.1039/c5nj01047k
Comment 4:The “Self-assembly mechanism of the graphene-like monolayer film” needs to be carefully polished. There are several misleading discussions. (1) Ferrous ions do not bind a “hydroxide radical”. Hydroxide radical is not a scientific name, and it should be hydroxyl radical. (2) Hydroxide radical may oxide Fe2+. (3) The authors should provide references to the claimed mechanism and also consider the effects from Fe3+.
Response: Thank you for the advice. We have revised this. (Line 159-191)

Reviewer 3 Report
This work reports a CD synthesis method by a one-step hydrothermal method with citric acid (CA) as the carbon source, ethylenediamine (EDA)
as the passivator and FeSO4â–ª7H2O as the catalyst. The work can be consider to be published after proper revision.
- The introduction need s to be approved. The current difficulty of synthesis CD and the discussion of the state of the art should be provided to point out the main novelty and advances of this work.
- What is the synthesis mechanism? It is suggested to add a more clear discussion of the possible mechnism of the CD synthesis.
- The carbon materials characterization can be improved by adding measurement of Raman analysis and electrical conductivity measurement of the samples.
- For the electrochemical study, please discuss the stability of the materials.
Author Response
Dear Editor,
Thank you very much for your attention and the reviewers’ comments on our manuscript entitled “Research on preparation methods of carbon nanomaterials based on self-assembly of carbon quantum dots” (ID: molecules-1565409). The comments are all valuable and very helpful for revising and improving our manuscript, as well as the important guiding significance to our researches.
We have checked and revised our manuscript carefully according to these comments, and the revised portions are marked in red in the manuscript. Enclosed please find the responses to the reviewers. We sincerely hope this manuscript will be finally acceptable to be published on Molecules.
Thank you and best regards.
Responses to reviewer’s comments are as follows:
Thank you for your positive suggestions. Please allow us to answer your concerns below.
Comment 1: The introduction needs to be approved. The current difficulty of synthesis CDs and the discussion of the state of the art should be provided to point out the main novelty and advances of this work.
Response: Thank you for your recognition.
Comment 2:What is the synthesis mechanism? It is suggested to add a more clear discussion of the possible mechnism of the CD synthesis.
Response: Thank you for the advice. (1) The synthesis mechanism of CDs with citric acid (CA) as the carbon source and ethylenediamine (EDA) as the passivator can be found in the introduction. (Line 50-60) (2) The synthesis mechanism of CDs with citric acid (CA) as the carbon source, ethylenediamine (EDA) as the passivator and FeSO4â–ª7H2O as the pre-catalyst can be found in the Self-assembly mechanism of the graphene-like monolayer film. (Line 166-171) (3) We found literature supporting that both divalent and trivalent iron ions can catalyze amide bond formation between phenylacetic acid and aniline. (Line 445-447)
[23]Basavaprabhu, B.; Muniyappa, K.; Panguluri, N. R.; Veladi, P.; Sureshbabu, V. V. A simple and greener approach for the amide bond formation employing FeCl3 as a catalyst. New. J. Chem. 2015, 39 (10), 7746-7749. DOI: 10.1039/c5nj01047k
Comment 3: The carbon materials characterization can be improved by adding measurement of Raman analysis and electrical conductivity measurement of the samples.
Response: Thank you for the advice. Due to the epidemic, the time of our return to school cannot be determined, and there is no way to make up for the experiment at present.
Comment 3: For the electrochemical study, please discuss the stability of the materials.
Response: Thank you for the advice. We have supplemented the figure and discussion of stability. (Line 154-156, 537,537, 574,575)
